# Application of social network analysis in transportation network based on AIS data

Pengfei Ouyang
Collaborative Innovation Center
for Transport Studies
Dalian Maritime University
Dalian, China
1002635287@qq.com

Yi Zuo
Navigation College
Dalian Maritime University
Dalian, China
zuo@dlmu.edu.cn

Junhao Jiang
Navigation College
Dalian Maritime University
Dalian, China
dmujiangjunhao@126.com

Peng Jia
Collaborative Innovation Center
for Transport Studies
Dalian Maritime University
Dalian, China
jiapeng@dlmu.edu.cn

*Abstract*—This article aims to explore ways to improve the efficiency and management level of maritime transportation by studying the key nodes in the Bohai Bay maritime transportation network and their impact on the network structure. Using the container ship data of the Bohai Bay in the first half of 2018, the Bohai Bay is divided into a 100×100 grid, with the ship trajectory points in the grid as nodes, the ship tracks as edges, and the frequency of the tracks as the weight of the edges. Build a network. The community detection method is used to identify the top four communities with the largest number of nodes, and the z-P matrix is used to divide the nodes in each community into different roles. In the experiment, the top 10 nodes in weighted degree, closeness centrality, and betweenness centrality, as well as provincial hubs and connector hubs were deleted respectively, and the average clustering coefficient and average path length of the community and the entire network before and after deletion of key nodes were analyzed. The change. The study found that the impact of deletion of key nodes on different communities and the entire network is significantly different, providing theoretical support and practical guidance for maritime traffic management.

*Keywords—Social network analysis, maritime transportation network, key nodes, network topology properties*

## I. INTRODUCTION

Maritime transportation networks are a vital component of the global trade and logistics system, responsible for transporting goods across the world. About 80% of global trade volume and more than 70% of global trade volume are completed through maritime transportation networks and transported to ports around the world for processing [1]. Research on maritime transportation networks faces many challenges, including data complexity, dynamic changes in the network, and the need for multi-scale analysis. The continuous advancement of data analysis and network science methods presents vast potential for studying maritime transportation networks.

Xu et al. studied the global liner shipping network structure and proposed a new concept of "gateway-hub" ports, which provided new insights into understanding the structural organizational complexity of GLSN and its correlation with international trade [2]. Using ports as nodes and routes between ports as edges, César Ducruet used graph theory and complex network methods to analyze the impact of the global financial crisis and the COVID-19 epidemic on the shipping network, and study the network's resilience [3]. Mou et al. studied the impact of China's coastal port disruptions on maritime network reliability, especially the assessment of the importance and risk of typhoons to ports[4]. Hoshi Tagawa et al. used network analysis to explore how cooperation affects the competitiveness and hierarchy of ports[5]. Liu et al. used the range of node connections, connection time and service capacity in graph theory to study the resilience of European port networks[6]. With the development and popularization of AIS technology, a large amount of maritime traffic data can be collected and analyzed in real time. Researchers have begun to pay attention to important information such as traffic flow, navigation trajectory, ship behavior, etc., and study the structure and dynamics of the maritime traffic network. Behavior, it is widely used in maritime supervision, traffic flow analysis, ship collision avoidance and other fields [7].Yan et al. proposed to convert ship trajectories with rich position information into ship trip semantic objects (STSO) with semantic information, and further integrated STSO into the nodes and edges of directed maritime traffic graphs for application Ship trajectory prediction, abnormal behavior detection and maritime traffic network evolution analysis, etc. [8]. Sui proposed to use ships as vertices and the relationships between ships as edges to construct a complex network of maritime traffic conditions and study its evolution[9]. In the article Maritime traffic network extraction and application based on AIS data, the ship trajectories in the AIS data are grouped by clustering method, and graph theory is used to abstract the traffic nodes and edges, and a ship abnormal behavior based on kernel density estimation is proposed. detection method[10]. Mou et al. used the DBSCAN algorithm to extract feature areas and boundary contours from trajectory points to form nodes, and constructed edges through frequent trajectory pattern recognition to study water route network extraction methods[11].

The Bohai Bay is an important maritime transportation hub, connecting the two major economic zones of North China and Northeast China. The surrounding ports have huge throughput and dense ships. Its trajectory information contains rich transportation network characteristics. Identifying the key nodes in the region is crucial to the actual maritime situation of the Bohai Bay. Traffic management has important implications. This paper divides the Bohai Bay into a 100×100 grid, uses the ship trajectory points in the grid as nodes, the track as the edge,

and the track frequency as the weight of the edge to construct a network to study the following issues:

1.Identification method of key nodes: Use the z-P matrix to divide the nodes into provincial hubs, connector hubs, kinless hubs, ultra-peripheral nodes, peripheral nodes, non-hub connector nodes and non-hub kinless nodes. Center-independent nodes[12]; and use the network topology properties to find the top ten nodes in weighted degree, closeness centrality, and betweenness centrality, and compare them with the provincial hubs and connector hubs are defined as key nodes.

2.The sensitivity of different communities to the deletion of key nodes: Select the first four communities with the largest number of nodes in the network, analyze the average clustering coefficient and average path length changes of each community before and after deleting key nodes, and compare the sensitivity of different communities to the deletion of key nodes.

3. The impact of key node deletion on the network structure: Analyze the changes in the average clustering coefficient and average path length of the entire network before and after deleting the key nodes of the community with the largest number of the first four nodes, and understand the role of key nodes and the impact of their deletion on the network. Impact and how it can serve as a transshipment and exchange point to improve network transportation efficiency and economy[13], providing theoretical support and practical guidance for maritime traffic management.

This article consists of four parts. The first part explains the research background and research status; the second part is the experimental design, explaining the content and methods of each step of the experiment; the third part is numerical results, selecting data from the Bohai Bay for analysis; the last part is conclusion

## II. EXPERIMENTAL DESIGN

Container ships are the core tool of modern global trade, responsible for about 90% of the maritime cargo transportation in global trade volume, involving ports and routes around the world, and their networks are complex and highly interconnected. Therefore, this article will use Bohai Bay container ship data in the first half of 2018 as the research object.

### A. Application of Social Network Analysis

*1) Definition of Nodes:* This study divides the Bohai Bay area into a 100×100 grid, as shown in Figure 1. For each grid, if there is a ship trajectory point in the grid, the ship trajectory point is used as a node; For grids with two or more trajectory points, the average location (longitude and latitude) is used; if the grid If there are no ship track points in the grid, the grid will be deleted.

*2) Definition of Links:* Figure 2 shows the construction logic of the adjacency matrix. Group the data by MMSI; for each MMSI, in chronological order, if the ship moves from this grid to another grid, edges are connected between the two grids. Specifically, if A reaches C via B, then A must be connected not only to B, but also to C, and so on.

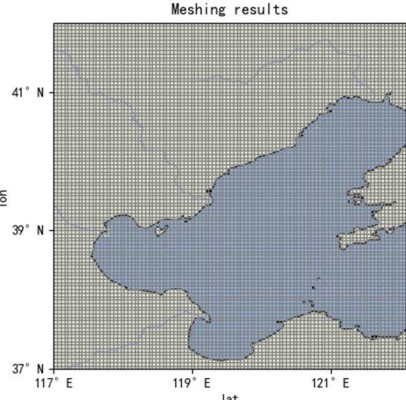

Fig. 1.   Meshing results.

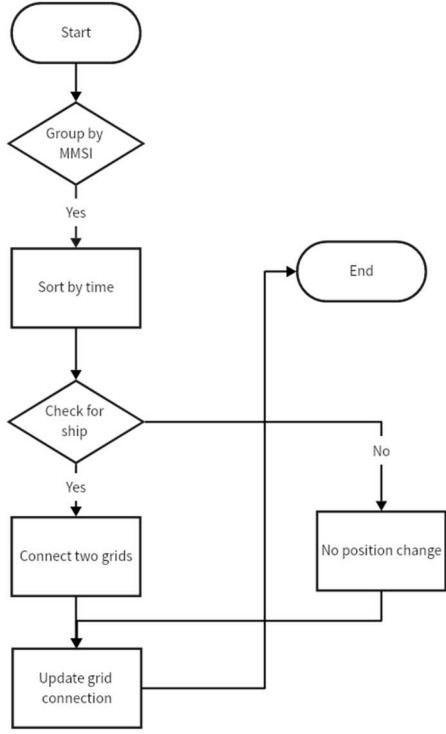

Fig. 2.   the construction logic of the adjacency matrix.

Figure 3 shows the construction logic of the weight matrix. Group the data by MMSI; if each MMSI passes through the same grid pair once, the weight increases; finally, the weights are normalized.

### B. Community Detection

There are many algorithms for community detection. Santo Fortunato once introduced a variety of methods for community detection, especially the application of techniques designed by statistical physicists in actual networks[14]. Zhao et al. used the Louvain algorithm, Girvan-Newman Algorithms and other community discovery technologies have successfully identified key sea areas and navigation nodes[15]. Li et al. proposed a new community detection method, the modified Medoid-Shift method, which combines the principles of Medoid-Shift and K-

Nearest Neighbors to improve the performance of community detection[16].

During the experiment, this article will use the community detection algorithm in Gephi software to randomly divide the network into different communities, and select the top four communities with the largest number of nodes for research.

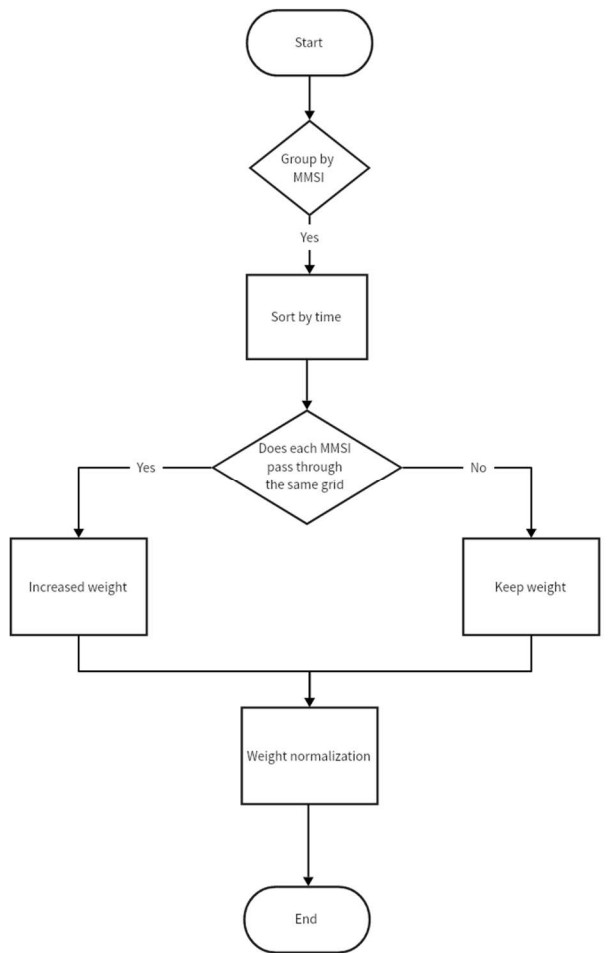

Fig. 3. the construction logic of the weight matrix.

## C. Network Centrality

In network science, weighted degree, closeness centrality, and betweenness centrality are common indicators used to measure the importance and characteristics of network nodes.

Weighted degree: Reflects the strength of node connections in the network.

Closeness centrality: High closeness means that the average distance between that node and other nodes is shorter.

Betweenness centrality: High betweenness nodes play an important "intermediary" role in the network, controlling the flow of information or resources in the network.

In addition, according to the different z and P values in the z-P matrix, this article can divide the nodes into the following seven different roles:

z values: measures how 'well-connected' node i is to other nodes in the module. High values of z indicate high within-module degrees and vice versa.

P values: measures how 'well-distributed' the links of node i are among different modules. The participation coefficient P is close to 1 if its links are uniformly distributed among all the modules, and 0 if all its links are within its own module.

1.When z≥2.5,

Provincial hubs: P≤0.3

Connector hubs: 0.3＜P≤0.75

Kinless hubs: P＞0.75

2.When z＜2.5,

Ultra-peripheral nodes: P≤0.05

Peripheral nodes: 0.05＜P≤0.62

Non-hub connector nodes: 0.62＜P≤0.8

Non-hub kinless nodes: P＞0.8.

The calculation formulas for z and P values in the matrix are as follows:

$$z_i = \frac{K_i - \overline{K_{s_i}}}{\sigma_{K_{s_i}}} \qquad (1)$$

where $K_i$ is the number of links of node i to other nodes in its module $s_i$, $K_{s_i}$ is the average of K over all the nodes in $s_i$, and $\sigma_{K_{s_i}}$ is the standard deviation of K in $s_i$.

$$P_i = 1 - \sum_{s=1}^{N_M} \left(\frac{K_{is}}{k_i}\right)^2 \qquad (2)$$

where $K_{is}$ is the number of links of node i to nodes in module s, and $k_i$ is the total degree of node i, $N_M$ is the number of communities.

## III. NUMERICAL RESULTS

First, the AIS data of container ships in the first half of 2018 was denoised: each MMSI was separated separately, and a total of 74 MMSI AIS data were obtained. After deleting the data of the ship track only in a certain small area, a total of 66 pieces were left. For the AIS data of MMSI, 66 pieces of AIS data were individually denoised and then summarized to obtain the noise-reduced data of container ships in the first half of 2018. The scatter plot of ship trajectories after noise reduction is shown in Figure 4.

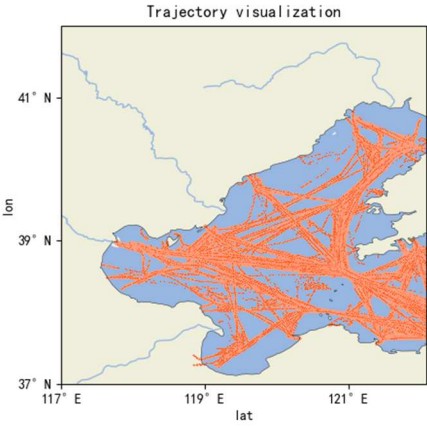

Fig. 4. Scatter plot of ship trajectory after noise reduction.

## A. Construction of Transportation Network

Divide the Bohai Bay into a 100×100 grid, take the trajectory points in the grid as nodes, determine whether there are edges between the grids based on whether there are ship tracks, and then increase the grid every time the ship passes through the same grid pair. The weight of the edge between the grid pairs is finally normalized, and the Bohai Bay maritime container ship traffic network can be obtained, as shown in Figure 5. After arranging the nodes by longitude and latitude, we can see the network and the trajectory of the ship after noise reduction. The scatter plots are generally similar, and the network restores the position information of the ship well.

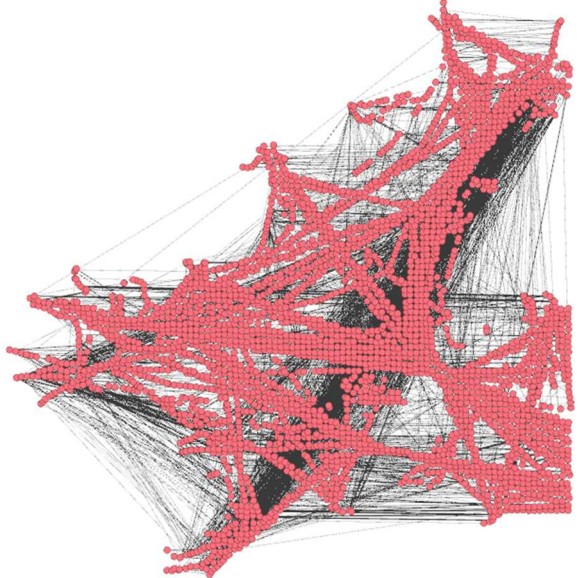

Fig. 5. Network Diagram.

## B. Analysis of Transportation Network

Using Gephi software to perform topological analysis on the network, various topological data of the network can be obtained as shown in Table I.

TABLE I.          NETWORK TOPOLOGY DATA

| Weighted Degree | Network Diameter | Graph Density | Average Clustering Coefficient | Average Path Length |
|---|---|---|---|---|
| 0.006 | 31 | 0.002 | 0.283 | 8.248 |

Humphries et al. introduced a small-world coefficient called $\sigma$ to describe the characteristics of small-world networks[17], and its formula is:

$$\sigma = \frac{C/C_{rand}}{L/L_{rand}} \qquad (3)$$

where $C$ is the average clustering coefficient of the actual network, $L$ is the average path length of the actual network, $C_{rand}$ and $L_{rand}$ are the average clustering coefficient and average path length of the equivalent random network with the same node degree distribution respectively.

On this basis, Telesford et al. proposed a new small-world coefficient called $\omega$[18], and its formula is:

$$\omega = \frac{L}{L_{rand}} - \frac{C}{C_{latt}} \qquad (4)$$

where among them, $C_{latt}$ is the average clustering coefficient of the equivalent lattice network.

According to the above two formulas, it can be concluded that the $\sigma$ coefficient of the network is 1.9 and the $\omega$ coefficient is 0.003. The network is a small-world network.

## C. Community in Transportation Network

Figure 6 shows the community detection results. It can be found that the network has a total of 28 communities, of which 4 communities have more than 200 nodes, namely community 13, community 19, community 23, and community 26. The number of nodes in community 23 is The most is over 500. The network diagram after coloring the network by community is shown in Figure 7. The modularity of the network is 0.842. Therefore, this article selects the nodes of community 13, community 19, community 23 and community 26 for experiments.

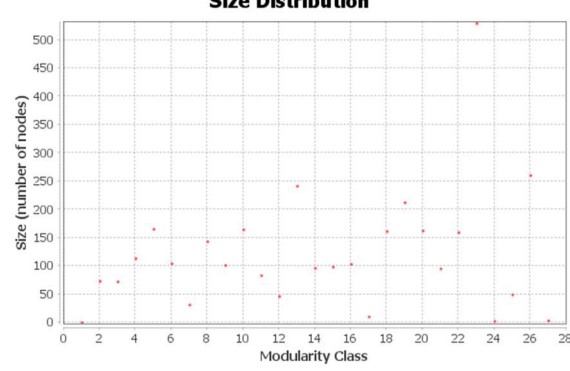

Fig. 6. Community detection results.

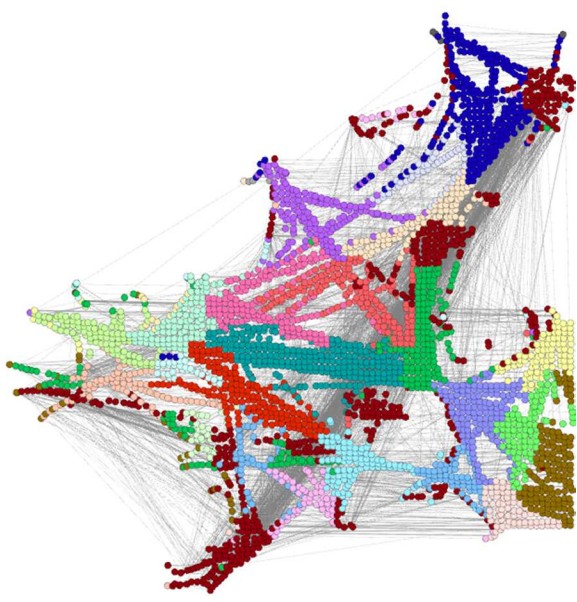

Fig. 7.   Community coloring network diagram.

### D. Comparison and Discussion

In order to facilitate research, this article defines the following nodes as key nodes, deletes them one by one, and analyzes the changes in the average clustering coefficient and average path length of the community and network before and after deletion. The key nodes are as follows:

1.The top 10 nodes with weighted degree in the community.

2.The top 10 nodes with closeness in the community.

3.The top 10 nodes with betweenness in the community.

4.Provincial hubs and connector hubs in the community

As shown in Figures 8 to 11, they are the node role distribution diagrams of community 13, community 19, community 23 and community 26 respectively. It can be seen that there are 3 provincial hubs and 1 connector hub in community 13; there are 2 provincial hubs and 3 connector hubs in community 19; there are 1 provincial hubs and 3 connector hubs in community 23; there is 2 provincial hubs and 5 connector hubs in community 26.

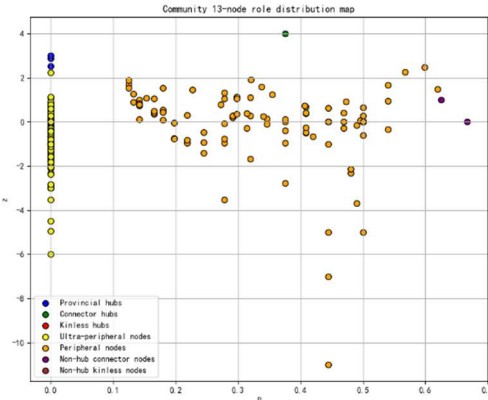

Fig. 8.   Community 13-node role distribution map.

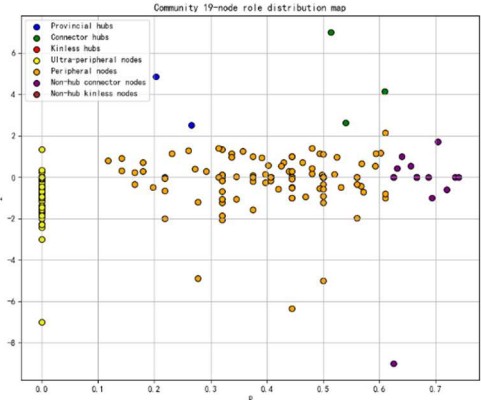

Fig. 9.   Community 19-node role distribution map.

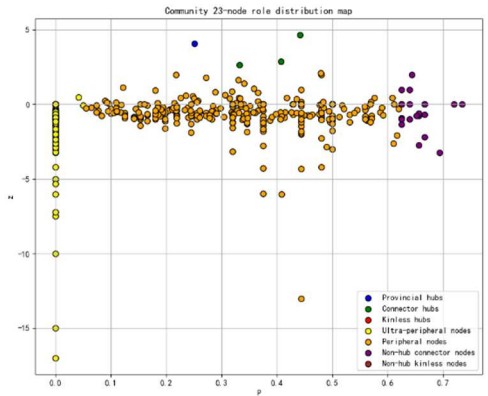

Fig. 10. Community 23-node role distribution map.

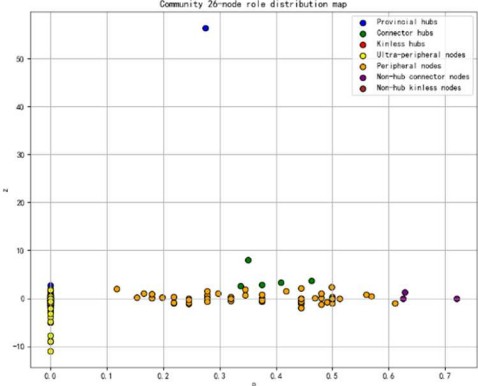

Fig. 11. Community 26-node role distribution map.

*1) Comparison in Communities:* Tables II to V show the average clustering coefficient and average path length of

communities 13, 19, 23 and 26 respectively before and after deleting key nodes.

For convenience, this article abbreviates the weighted degree as D, the closeness centrality as C, the betweenness centrality as B, the provincial hubs and the connector hubs as H, the average clustering coefficient as CC, the average path length as PL, and the original data as O.

In community 13, deleting the provincial hubs and the connector hubs has the greatest impact on the community, with its average clustering coefficient increasing by 0.1 and its average path length decreasing by 0.893.

In community 19, deleting nodes with the top 10 weighted degrees has the greatest impact on the community, with the average clustering coefficient decreasing by 0.025 and the average path length increasing by 0.968.

In community 23, deleting nodes in the top 10 weighted degrees and deleting nodes close to the top 10 in closness centrality both have a greater impact on the community. The average clustering coefficient of the former decreased by 0.049, and the average path length increased by 0.4; the average clustering coefficient of the latter decreased by 0.048, and the average path length increased by 0.433.

In community 26, deleting nodes with the top 10 weighted degrees and deleting provincial hubs and connector hubs have a more significant impact on the average path length of the community. The average path length of the former increased by 1.02; the average path length of the latter decreased by 1.402.

The sensitivity of community 23 to the deletion of key nodes is significantly lower than that of other communities. This may be because community 23 has more nodes than other communities and is the community with the largest number of nodes; while the sensitivity of community 26 is significantly higher than that of other communities.

The deletion of the top 10 nodes with weighted degree has a greater impact on the two communities of community 19 and community 23 than other key nodes.

The deletion of the provincial hubs and the connector hubs will have a greater impact on the two communities of community 13 and community 26 than other key nodes.

TABLE II.    THE AVERAGE CLUSTERING COEFFICIENT AND AVERAGE PATH LENGTH OF COMMUNITY 13 BEFORE AND AFTER DELETING KEY NODES

|  | O | D | C | B | H |
|---|---|---|---|---|---|
| CC | 0.304 | 0.303 | 0.304 | 0.311 | 0.314 |
| PL | 6.359 | 6.628 | 6.433 | 6.69 | 5.466 |

TABLE III.    THE AVERAGE CLUSTERING COEFFICIENT AND AVERAGE PATH LENGTH OF COMMUNITY 19 BEFORE AND AFTER DELETING KEY NODES

|  | O | D | C | B | H |
|---|---|---|---|---|---|
| CC | 0.314 | 0.289 | 0.311 | 0.293 | 0.309 |
| PL | 7.311 | 8.279 | 7.893 | 6.416 | 7.405 |

TABLE IV.    THE AVERAGE CLUSTERING COEFFICIENT AND AVERAGE PATH LENGTH OF COMMUNITY 23 BEFORE AND AFTER DELETING KEY NODES

|  | O | D | C | B | H |
|---|---|---|---|---|---|
| CC | 0.277 | 0.228 | 0.229 | 0.242 | 0.27 |
| PL | 3.176 | 3.576 | 3.609 | 3.46 | 3.23 |

TABLE V.    THE AVERAGE CLUSTERING COEFFICIENT AND AVERAGE PATH LENGTH OF COMMUNITY 26 BEFORE AND AFTER DELETING KEY NODES

|  | O | D | C | B | H |
|---|---|---|---|---|---|
| CC | 0.267 | 0.262 | 0.255 | 0.264 | 0.271 |
| PL | 7.939 | 8.959 | 8.014 | 8.281 | 6.537 |

*2) Comparison in Grids:* Tables VI to IX respectively show the average clustering coefficient and average path length of the entire network before and after the key nodes are deleted in communities 13, 19, 23 and 26.

In community 13, deleting the top 10 nodes with betweenness centrality has a greater impact on the network, with the average path length increasing by 0.178.

In community 19, deleting the top 10 nodes with betweenness centrality has a greater impact on the network, with the average path length increasing by 0.168.

In community 23, deleting the top 10 nodes with betweenness centrality and deleting nodes in the top 10 closness centrality have a greater impact on the network. The average path length of the former increased by 0.402, and the average path length of the latter increased by 0.356.

In community 26, deleting the top 10 nodes with betweenness centrality has a greater impact on the network, with the average path length increasing by 0.19.

The network is significantly more sensitive to the deletion of key nodes in community 23 than in other communities.

The network is significantly more sensitive to the deletion of the top 10 nodes with betweenness centrality than other key nodes.

Key node deletion has minimal impact on the network's average clustering coefficient, but generally results in an increase in average path length.

TABLE VI.    THE AVERAGE CLUSTERING COEFFICIENT AND AVERAGE PATH LENGTH OF THE NETWORK BEFORE AND AFTER DELETING KEY NODES IN COMMUNITY 13

|  | O | D | C | B | H |
|---|---|---|---|---|---|
| CC | 0.283 | 0.283 | 0.283 | 0.283 | 0.283 |
| PL | 8.248 | 8.311 | 8.366 | 8.426 | 8.227 |

TABLE VII. THE AVERAGE CLUSTERING COEFFICIENT AND AVERAGE PATH LENGTH OF THE NETWORK BEFORE AND AFTER DELETING KEY NODES IN COMMUNITY 19

|  | O | D | C | B | H |
|---|---|---|---|---|---|
| **CC** | 0.283 | 0.281 | 0.282 | 0.282 | 0.283 |
| **PL** | 8.248 | 8.306 | 8.309 | 8.416 | 8.256 |

TABLE VIII. THE AVERAGE CLUSTERING COEFFICIENT AND AVERAGE PATH LENGTH OF THE NETWORK BEFORE AND AFTER DELETING KEY NODES IN COMMUNITY 23

|  | O | D | C | B | H |
|---|---|---|---|---|---|
| **CC** | 0.283 | 0.282 | 0.277 | 0.28 | 0.282 |
| **PL** | 8.248 | 8.38 | 8.604 | 8.65 | 8.325 |

TABLE IX. THE AVERAGE CLUSTERING COEFFICIENT AND AVERAGE PATH LENGTH OF THE NETWORK BEFORE AND AFTER DELETING KEY NODES IN COMMUNITY 26

|  | O | D | C | B | H |
|---|---|---|---|---|---|
| **CC** | 0.283 | 0.282 | 0.283 | 0.283 | 0.283 |
| **PL** | 8.248 | 8.28 | 8.368 | 8.438 | 8.338 |

*E. Analysis and Discussion*

Through the comparison and discussion of data in the previous experiments, the following phenomena can be found:

1. Nodes with high betweenness centrality, provincial hubs and connector hubs have a greater impact on the community.

2. The deletion of provincial hubs and connector hubs in communities 13 and 26 reduces the average path length of the community, improves network connectivity, and increases propagation efficiency; the average community clustering coefficient increases and local connectivity is enhanced. , local connectivity number, local propagation efficiency becomes higher.

3. Community 23 shows significantly lower sensitivity to key node deletion compared to other communities, while the sensitivity of community 26 is significantly higher than that of other communities.

4. Nodes with high betweenness centrality have a greater impact on the network, and the average clustering coefficient of the network is basically not affected by key nodes.

5. The key nodes of community 23 have a greater impact on the network.

When the provincial hubs and connector hubs of community 13 are deleted, the average path length of the network is also reduced, which means that ships can try to avoid this sea area when sailing, as shown in Figure 12, thus improving transportation efficiency.

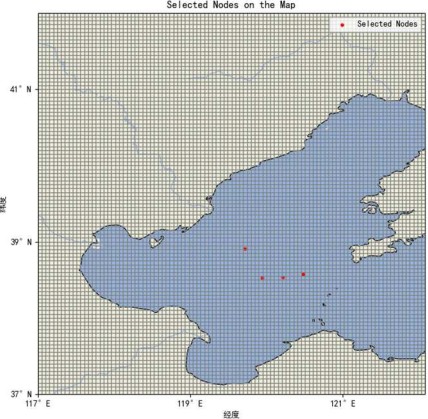

Fig. 12. Community 13 provincial hubs and connector hubs.

When nodes with high betweenness centrality and nodes with high closeness centrality are deleted in community 23, the average path length of the network increases significantly and the connectivity becomes worse, which means that the supervision of these sea areas should be strengthened, as shown in Figure 13. Try to avoid these sea areas becoming impassable due to emergencies and reducing transportation efficiency.

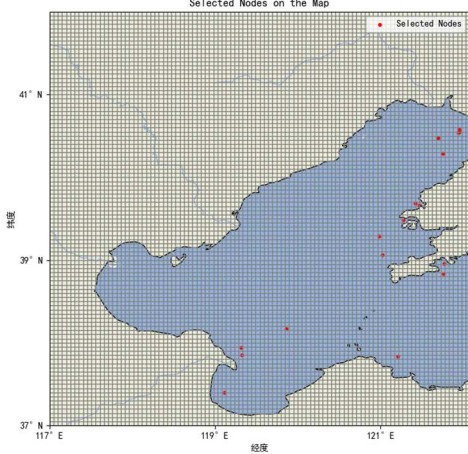

Fig. 13. Community 23 betweennesscentrality and closenesscentrality top 10 nodes.

## IV. CONCLUSION

This study conducted a comprehensive analysis of the Bohai Bay maritime transportation network and identified key nodes and communities in the network by constructing a network model based on AIS data. The results reveal differences in the sensitivity of different communities to the deletion of key nodes, as well as the impact of key nodes on network structure and function. In particular, community 23 shows lower sensitivity to the deletion of key nodes, while community 26 shows higher sensitivity. In addition, the deletion of nodes with high closeness centrality and provincial hubs as well as connector hubs has a greater impact on communities and networks, indicating that

these nodes play a key role in connectivity and stability in the network.

The study also found that the deletion of key nodes has little impact on the average clustering coefficient of the network, but usually leads to an increase in the average path length, thereby affecting the connectivity of the network. In particular, through the analysis of community 13, we learned that the deletion of provincial hubs and connector hubs in community 13 can improve network connectivity and local propagation efficiency.

The study offers theoretical support and practical guidance for maritime traffic management, highlighting the importance of protecting key nodes and optimizing network performance. In actual maritime traffic management, the monitoring and protection of these key nodes should be strengthened to avoid the impact of emergencies on transportation efficiency.

## ACKNOWLEDGEMENT

This work was supported in part by the National Natural Science Foundation of China (grant nos. 52131101 and 51939001), the Liao Ning Revitalization Talents Program (grant no. XLYC1807046), and the Science and Technology Fund for Distinguished Young Scholars of Dalian (grant no. 2021RJ08).

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
