# OpenReview forum: "Application of social network analysis in transportation network based on AIS data"
_IEEE.org/ICIST/2024/Conference — IEEE ICIST 2024 Conference Submission_

### Official Review · Reviewer_CvJJ · 2024-08-22
**This article is well written and can be accepted.**

**Rating:** 8
**Confidence:** 3

**Review:**

1.The engineering background of the proposed problem should be stated more clearly to readers. The literature review is insufficient. Some recently published papers should be included in the references list. 2.There are many grammatical and typographical errors in the manuscript. Please check the full text carefully and correct them. 3. Section 3 contains mathematical equations. It is suggested to include the proper explanation along with the equations. 4.In simulation section, more analysis and descriptions should be given to show the effectiveness of the developed method.

---

### Official Review · Reviewer_1REh · 2024-09-03
**Application of social network analysis in transportation network based on AIS data**

**Rating:** 7
**Confidence:** 4

**Review:**

1. Please improve the equations by adding brief insights about them.

2. Each reference has to be double verified, and also reference writing style needs
to be uniform. It is essential to review all references, fill in any gaps with Volumes, Issues,
and Pages, and revise any inaccurate information. Further, for current references that aren’t
yet listed in Volume and Issue, DOI numbers must also be added.

3. How to set the parameters of proposed method for better performance?

---

### Decision · Program_Chairs · 2024-09-06

Accept (Oral)